**Biological science practices**  

ecology

career, currency, data contributions

**Author for correspondence:**
Mark Westoby
e-mail: mark.westoby@mq.edu.au

# Motivating data contributions via a distinct career currency

Mark Westoby[1], Daniel S. Falster[2] and Julian Schrader[1,3]

[1]Department of Biological Sciences, Macquarie University, Sydney, NSW 2109, Australia
[2]School of Biological, Earth and Environmental Sciences, University of New South Wales, Sydney, NSW 1466, Australia
[3]Department of Biodiversity, Macroecology and Biogeography, University of Goettingen, Goettingen, Germany

MW, 0000-0001-7690-4530; DSF, 0000-0002-9814-092X; JS, 0000-0002-8392-211X

If collecting research data is perceived as poorly rewarded compared to data synthesis and analysis, this can slow overall research progress via two effects. People who have already collected data may be slow to make it openly accessible. Also, researchers may reallocate effort from collecting fresh data to synthesizing and analysing data already accessible. Here, we advocate for a second career currency in the form of data contributions statements embedded within applications for jobs, promotions and research grants. This workable step forward would provide for peer opinion to operate across thousands of selection and promotion committees and granting panels. In this way, fair valuation of data contributions relative to publications could emerge.

## 1. Introduction: the data-flow problem and motivations

Ideally in a research community, all new data would be made promptly open for anyone with the knowledge and imagination to interpret it. Also ideally, a flow of fresh and useful data would meet the needs of data interpreters who have ideas to test. Currently, incentives or motivations in the research community are not producing this ideal balance of activities (e.g. [1,2]). This essay discusses why the research community falls short of the ideal, and makes a simple and workable proposal to improve the situation.

Behaviours at two levels control the flow of research data. At one level, researchers who have collected data make decisions about how much of it to make openly available at the time of publication, or perhaps even before publication. At another level, people engaged in research make decisions about how much of their time to allocate to collecting fresh and useful data, versus how much to allocate to analysing and interpreting data already available. The two levels of behaviour are linked, because if people feel their careers will not progress by contributing data to narratives led by others, they can be expected to shift their efforts away from producing new data and towards analysis and interpretation.

Our argument proceeds as follows. First, we discuss how existing currencies for career progress do not motivate collecting fresh data as strongly as is widely believed desirable. Second, making one's data openly accessible is shown to have the form of a public goods game (PGG). The PGG framework is shown to explain both the existing motivations for making data public, and also the limitations on those motivations. Third, it is suggested that a 'data contributions' currency should be established separately from contributions via publication. Examples are given of what a data contributions statement might contain, and the question of its valuation relative to publications is addressed.

## 2. Motivation via publications

At present, the principal currency for career progress in scientific research is authorship of peer-reviewed publications, together with citation impact of

those publications. This is true for winning postdocs and tenure-track positions, for promotions, for applications for grant funds, and for recognition via prizes and awards. Currencies can also be thought of as motivations.

Under existing guidelines (box 1), data contributions are only one of 4–5 criteria for being included as an author on publications. These guidelines focus (understandably) on authors accepting responsibility for the conclusions drawn by the paper. Currently, these guidelines are being considerably stretched. Logan et al. [3] estimated that for ecology in 2010, 78% had some level of non-compliance with protocols from International Council of Medical Journal Editors and 48% with protocols from Ecological Society of America. Many data synthesis papers are being published with 10 or more authors (e.g.[ 4–7]). The length of the authorship list reflects a desire to recognize data contributions. It is not credible that all of those authors were serious participants in each of conception, execution and analysis. An extreme recent example is a paper [8] with 729 authors including ourselves who have contributed to the TRY plant trait database (www.try-db.org).

Among other indications that the publication and citation system is not perceived as rewarding data contributions sufficiently, journals have emerged that specialize in publishing 'data papers', publications that contain no narrative or conclusion but the only description of a dataset.

With the developing practice of citing to data via a DOI, it has been hoped that citations to DOIs would become an effective index of data contributions, comparable to citation impact for authorship [17]. Pierce et al. [18], reporting from a workshop, proposed an integrated system for tracking data use via persistent identifiers (PIDs) for each of datasets (DOIs), researchers (ORCID numbers) and publications (DOIs). Persistence means that if a publication aggregates different data sources, subsequent use will not just cite the aggregating publication but will connect back to the original sources. If identifiers for all three of datasets, researchers and publications came to be routinely attached and linked in this way, it ought to become possible to track and accredit data use quantitatively and automatically. Pierce et al. hoped that this would feed back through a 'virtuous cycle' to encourage data collection.

If persistent identifiers came to be implemented effectively, it is possible they might help. Our opinion, though, is that recognition via publication and citation is not a motivation that is ever likely to fully achieve the optimal balance between data collection and data interpretation. The path forward that has best prospects, we suggest, is to create a separate currency through which data collection can be valued.

## 3. Open data access as a public goods game

Open data [19–24] is when research data are made publicly accessible at least at the time of first publication. Indeed, ideally they should be made accessible independently of publication, to avoid the file-drawer effect [24]. Advocacy for open data has included a Berlin Declaration (https://openaccess.mpg.de/Berlin-Declaration), a Bouchout Declaration (http://www.bouchoutdeclaration.org/declaration/) and a Denton Declaration (https://openaccess.unt.edu/denton-declaration). Guidelines have been developed to encourage data sharing under the acronym FAIR, for findable, accessible, interoperable and reusable [25]. In related tendencies over the past 15 years,

**Box 1.** Summary of two existing protocols for what justifies authorship of research papers.

The CRediT (https://casrai.org/credit/) contributor role taxonomy [9] was designed to record authorship contributions more precisely and consistently. It has been adopted by at least 120 journals [10]. Extensions have been suggested to contributors other than authors [11,12]. Formulae have been proposed for partitioning credit instead of just attributing citations equally to all authors [13,14]. Two of the 14 roles, investigation and data curation, correspond to data contributions. Although one purpose of CRediT is to discourage authorships that are seen as undeserved, CRediT does not itself lay out rules for what justifies authorship. Two protocols that do propose such rules are

The International Committee of Medical Journal Editors (ICMJE) has developed the 'Vancouver Protocol' [15], which has been adopted by many journals and endorsed with slight rephrasing by McNutt et al. [9]. This recommends that authorship be based on the following criteria:

1. Substantial contributions to the conception or design of the work; or the acquisition, analysis, or interpretation of data for the work; AND
2. Drafting the work or revising it critically for important intellectual content; AND
3. Final approval of the version to be published; AND
4. Agreement to be accountable for all aspects of the work in ensuring that questions related to the accuracy or integrity of any part of the work are appropriately investigated and resolved.

The Ecological Society of America's Code of Ethics [16] recommends authorship should require at least three of

1. conceived the ideas or experimental design;
2. participated actively in execution of the study;
3. analysed and interpreted the data; OR
4. wrote the manuscript.

Similarly to the ICMJE guidelines, authors are expected to know which co-authors are responsible for other parts of the work and to have confidence in their integrity. Notice that data contributions cannot be recognized by authorship unless the data contributor agrees with the interpretation.

In summary, recognizing data contributions is not an important focus in these authorship protocols.

global-scale data syntheses with many authors have become more common (e.g. [26–29]), and several countries have set up centres that have the aim of organizing workshops to drive forward the synthesis of ecological data [30].

At first glance it might seem that open data access would always accelerate the communal aim of advancing science. Data would be available for synthesis and reanalysis sooner, and by a wider range of talents. But if the motivations bearing on researchers do not motivate people sufficiently to collect high-quality data, then the balance of effort as between collecting and interpreting data may not be what is best for research progress over the longer term.

**Figure 1.** Schematic summarizing how a basic public goods game operates (left side), how four mechanisms counteract the basic tendency towards zero data contribution into the public pool (across top), and how data contributions statements or indices can potentially support motivation to collect new data and to contribute it to the open pool (right side), depending on the valuation attached by peer panels responsible for competitive appointments and grants.

The open-access problem is an instance of a PGG. Framing the situation as a PGG can help us to understand more explicitly what different players stand to gain or lose, and also what features of the social and institutional setting may be more versus less conducive to data sharing.

In a basic PGG (figure 1, left side), players have a private endowment. In each round of the game, they contribute some fraction of their endowment to a pool, then the pool is multiplied by an increase factor, and then it is shared out among players. The motivation for each player is to maximize their endowment. If contributing to the pool is a binary choice, then those who contribute can be labelled cooperators and those who do not can be labelled defectors or free-riders. Alternatively, the amount of contribution can be modelled as a continuous variable.

In this basic PGG, each player does best to make no contribution, since whatever the share-out from the pool, net return to each player is greater when they have not spent any from their endowment. This basic PGG approximates human situations such as paying national taxes and receiving in return the use of a free public road system. Individuals will avoid paying taxes if they can, since they continue to be able to use the roads regardless. From the point of view of an individual taxpayer in a national programme with a large number of participants, the benefit from evading taxes is large while road deterioration due to their particular non-payment is small. And consequently, from the point of view of governments representing the community, enforcement is necessary to elicit payment of taxes.

Considering open data as a case of this basic PGG, individual researchers are making the decision whether (or how early) to put their data into the public domain. The short-term public good from data becoming widely accessible is clear. But when researchers put their data into the public domain, there is a chance that others will use them to publish interpretations they might themselves otherwise have published. Consequently, players will typically feel more confidence about obtaining publication credit when they retain data in their own hands [32].

This motivation not to contribute data is counteracted by four main mechanisms, each of which can be understood through the lens of public good games (figure 1, upper part). Three of the mechanisms currently important in supporting some level of cooperation in data sharing are predicted from theoretical PGG: (i) when interactions between researchers are iterated, transparent and local within a network, meaning that each researcher does not interact with all others but only with a subset they are able to identify; (ii) when there is nonlinearity in the benefit function; and (iii) external compulsion, for example by funding agencies or by journals; this can be thought of as attaching a cost to free-riding. There is also (iv) a cultural propensity to collaborate, not predicted by theoretical PGG but emerging consistently in experiments with real people.

Localness in conjunction with iteration has mainly been studied via the 'spatial evolutionary PGG' [33], where players interact not with the entire population but only with a subset to whom they are connected in a network. Under these circumstances, local clusters of collaboration may emerge and persist. Local collaboration is favoured when the PGG is transparent rather than opaque, meaning that players are aware of who is contributing how much. Transparency favours cooperation because it makes it possible for players to select cooperators rather than free-riders for future interactions [34]. Continuing cooperative data sharing with a limited number of collaborators, and with transparency, is the most common operating pattern currently within ecology. Data sharing is common among networks or working groups where the people have personal contact and a reasonable expectation of reciprocation.

Nonlinearity of response is illustrated by the volunteer's dilemma (e.g. [35]), where (in a simple form of the game) some fixed number of cooperators is needed to produce the public good. Think for example of amateur football matches, where each side needs at least 10–11 players plus maybe a coach for a credible match to be played. There is no benefit up to approximately 11 cooperators, then a sharp increase around about 11 cooperators, then limited and eventually

zero further increase in benefit as still more cooperators are added. One possible stable outcome in such a game is for no cooperation. But once an association has been formed that includes enough cooperators to generate the benefit, then each of them is inhibited from shifting to free-riding by the risk of strong losses if the number of cooperators falls below the number needed. Via this disincentive, such an association of cooperators can also be stable. This sort of nonlinearity applies to many situations in research. For publications in ecology making a claim to global coverage, there may be a minimum need for perhaps 8–10 laboratories to contribute data, while the credibility of the claim does not increase much beyond 40 or 50 laboratories.

In 'experimental economics' [36], real humans are asked to participate in trials having the structure of public goods games. In such experiments, players will typically begin in quite a cooperative fashion, making substantial contributions to the pool. If the game is iterated over repeated rounds, contributions tend to decline as players become aware there are free-riders taking advantage. Both these tendencies are recognizable when making data available for research. The research world is characterized both by an ethos for communal outcomes, and by intense competition for positions and research grants. The intense competition is not necessarily direct self-interest. For example, laboratory leaders may see it as their role to ensure that research students get maximum credit for data they have collected.

Compulsion has become widespread—many funding agencies and journals now require that data be made public along with papers, or within some period after the funded project ends, or sometimes at the time of data collection. But compulsion to make data available is far from ideal, for two reasons. First, compliance may be grudging and minimal (e.g. [37]). The data made public may cover the analyses actually reported in the paper, but may not (for example) provide all the individual replicates, nor other measurements or site properties that were not used in the publication, nor the unquantified but potentially helpful insight gained by the researchers in the course of collecting the data. Second (and in our opinion this is the really important deficiency of compulsory open access), the motivation for researchers to commit their time to collecting quality data has decreased, and the motivation to commit time to interpreting existing data has increased.

In summary, the public goods game framework helps us to understand why each of local iterated collaboration, nonlinear response, cooperative culture and external compulsion is having some effect in encouraging researchers to make their data public. But PGGs also help us to understand why the effects are limited, and none come close to the ideal that all data should become public at the time of collection. This brings us to the question of whether other steps might constructively affect motivations, besides recognition through data citation, and compulsion by journals and funding agencies.

## 4. Data contribution statements as a supplementary currency

The core problem is that authorship (still the principal currency for advancing research careers) is irremediably deficient as an indicator of data contributions. It is not really capable of discriminating volume or quality of data contributed, nor difficulties overcome in developing

---

**Box 2.** Elements that might be included in a data contributions statement.

Materials for a data contributions statement might include any of the following. The mixture would vary greatly depending on the individual.

1. Top 5 or 10 papers where a significant data contribution was made, each with 20-word explanation of the importance of the contribution.
2. Account of methods developed, that increased precision or convenience of measurement.
3. Account of difficult data contributed—from remote sites, or requiring much laboratory time—estimates of time spent might be provided as an indicator of the difficulty.
4. Intentional sampling of underrepresented taxa or locations.
5. Activities in data synthesis and integration [38].
6. Overall summary of volume of data entered into public domain.
7. DOIs for datasets contributed, plus citation counts to them [18].
8. Indices such as the data-index proposed by Hood & Sutherland [2], an equivalent for datasets of the h-index widely used for cites to publications.

---

measurement techniques. There is no consistency in the extent to which data contributors are included as authors. Lead authorship of data syntheses commonly falls to data compilers or interpreters rather than to primary contributors.

We propose that a practical step forward can be to establish or encourage a separate pathway to express data contributions. In the language of public goods games, this would be a second currency. What we suggest is that for job, promotion and grant applications, there should be distinct sections inviting researchers to describe what they believe their contributions are to data that are in the public domain. Box 2 suggests types of material that a data contributions section might contain.

In proposing free-form data contributions statements, we do not mean to discount the value of quantitative indices for data contributions such as items 6–8 in box 2. Given the invitation to make a data contributions statement, researchers will use whatever sort of evidence they think is most likely to persuade their peers on selection and promotion committees. That might very often include quantitative indices.

However, quantitative indices will have all the same attribution problems that have been discussed for authorship and citation indices. How to assess the contributions from project coordinators, postdocs, PhD students and professional staff? How highly should data compilation or curation be valued compared to fieldwork? What about long-term projects where successive generations of researchers have been involved? Also, any index that becomes widely used is prone to being gamed. A researcher writing a data contributions statement will be conscious that some of their peers think indices valuable, while others are cynical about them.

These limitations on quantitative indices can be offset by qualitative commentary, identifying the most important data contributions and the applicant's role in them (any of items 1–5 in box 2). People can be expected to write qualitative

comments in a way that puts a positive shading on their contributions. But they will be aware of a likelihood that their comments may be read by people who have independent knowledge about particular projects or datasets. This should moderate the self-promotion.

Our suggestion for a data contributions statement is similar to how publications and citations are often handled. Publication and citation counts and h-indices may typically be given in grant and job applications, but they are embedded in text that makes a case about the overall direction of research, and about particularly important or innovative papers and the applicant's personal contribution to them.

While we are not ourselves in a position to cause data contributions statements to become widespread, we note that this would be an easy thing for institutions to do. There is plenty of precedent for sections of this sort. For example, job applications commonly invite a section about teaching experience and philosophy; grant applications commonly include sections on translating research into application. It would be straightforward for granting agencies to provide space in application forms for a data contributions section, and similarly straightforward for universities to provide for it within job application or promotion forms.

There are recurring competitions that characterize a researcher's life, for jobs and grants and promotion. The competitors are assessed by selection committees consisting mainly of other members of the research community. These committees use their best judgement about what is most important. The result is that a disseminated community opinion operates, distributed across these very many assessments.

Our intention here is not necessarily to press for data contributions to be valued higher relative to interpretation, narrative and concepts. We do not (for example) propose that data contributions statements always count 20% towards the overall valuation of a grant proposal. We have often heard conversations along the lines that data contributions are undervalued. But equally, it is possible to argue that data only become valuable through answering a question or illustrating a concept, in other words through a publication with a narrative. In competitions for desirable positions, how often will people known for pioneering a measurement technology be appointed in preference to people known for concepts and for widely cited papers? We honestly do not know. Our point is that for the research community's judgement about this to be made in an explicit and open way, a data contributions currency needs to be widely visible, separately from a publication and citation currency. In that way, a communal valuation or market pricing for data contributions can emerge from the research community over time, via the peer judgement of innumerable selection and promotion committees and granting panels.

Data accessibility. This article has no additional data.

Authors' contribution. M.W. initiated the idea and wrote the manuscript in consultation with J.S. and D.S.F.

Competing interest. We declare we have no competing interests.

Funding. Fellowship funding was provided by Deutsche Forschungsgemeinschaft to J.S. (grant no. SCHR 1672/1-1), and by Australian Research Council to D.S.F. (FT160100113).

Acknowledgements. We thank Prof Mary O'Kane for helpful discussion.

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
