## [Peer Review File · Proceedings of the Royal Society B: Biological Sciences]

Review History

RSPB-2020-2830.R0 (Original submission)

Review form: Reviewer 1 (Alec Christie)

Recommendation

Accept with minor revision (please list in comments)

Scientific importance: Is the manuscript an original and important contribution to its field?

Good

General interest: Is the paper of sufficient general interest?

Excellent

Quality of the paper: Is the overall quality of the paper suitable?

Acceptable

Is the length of the paper justified?

Yes

Should the paper be seen by a specialist statistical reviewer?

No

Do you have any concerns about statistical analyses in this paper? If so, please specify them explicitly in your report.

No

It is a condition of publication that authors make their supporting data, code and materials available - either as supplementary material or hosted in an external repository. Please rate, if applicable, the supporting data on the following criteria.

Is it accessible?

N/A

Is it clear?

N/A

Is it adequate?

N/A

Do you have any ethical concerns with this paper?

No

Comments to the Author

Review - motivating data contributions via a distinct career currency

General comments

I think this is a useful contribution to the debate about how to value researcher contributions and I like how the authors link the issue to Public Goods Games. I like the examples given to demonstrate the principles of PGGs and how they apply to issues surrounding open data, but maybe a diagram or conceptual figure could help to bring the article alive a little more? In places the grammar and sentence structure needs to be cleared up and I've made some comments below.

I'd also suggest some greater engagement with the literature on valuing researcher contributions - the article just generally feels a little light on references. I'd point the authors to a recent preprint (I disclose by two researchers in my research group) that are proposing a new author-level metric that values impactful data and incentivises data sharing: 'The data-index: an author-level metric that values impactful data and incentivises data sharing' <https://doi.org/10.1101/2020.10.20.344226>. It would be interesting to discuss the strengths and limitations of such an approach here as the authors seem to err on the side of providing a qualitative assessment of data contributions in applications, rather than a quantitative metric to directly provide an alternative to the h-index and other authorship-related metrics.

Some more papers that could also be integrated are:

- 'Transparency in authors' contributions and responsibilities to promote integrity in scientific publication' <https://doi.org/10.1073/pnas.1715374115>

- 'Author contributions to ecological publications: What does it mean to be an author in modern ecological research?' <https://journals.plos.org/plosone/article?id=10.1371/journal.pone.0179956>

- 'Author sequence and credit for contributions in multiauthored publications' <https://journals.plos.org/plosbiology/article?id=10.1371/journal.pbio.0050018>

- 'Transparent Attribution of Contributions to Research: Aligning Guidelines to Real-Life Practices' <https://doi.org/10.3390/publications7020024>

- 'Attitudes to the rights and rewards for author contributions to repositories for teaching and learning' <https://doi.org/10.1080/09687760600837066>

- 'The need to quantify authors' relative intellectual contributions in a multi-author paper' <https://doi.org/10.1016/j.joi.2017.01.002>

- 'Is the "academic conservation scientist" becoming an endangered species?' <https://doi.org/10.1007/s13412-020-00633-6>.

Title

Title should be capitalised.

Abstract

The final sentence of the abstract (L21-24) has an erroneous or misplaced comma, it doesn't flow.

Introduction

L36 - This sentence is missing a word. Should it be "researchers who have collected data make decisions ABOUT how much of it to make openly available".

L38 - Same problem as for L36.

L35-43 - I wonder if there is any evidence or theories from psychology that can be used to strengthen this paragraph's argument?

2. motivation via publications

Subheading should be capitalised.

L59 - Should be "Currencies can ALSO be thought of as motivations."

L66 - ICMJE and ESA acronyms have not been defined yet (I know they are in Box 1 but that comes later). A reference to Box 1 should also be made so that people can understand what the protocols are if they don't know them.

L66-67 - 'Many data synthesis papers are being published with 10 or more authors'. Can you provide any evidence of this from the literature? Like you do on L138 for global data syntheses.

L70-72 - I agree! I myself have just published a paper where I included many co-authors purely for data contribution, and although I tried to get co-authors to contribute to the design and writing of the analysis, it's inevitable that some contribute a great deal and some contribute extremely little.

L71 - Provide a link to the database?

I don't see any mention of the CRediT (Contributor Roles Taxonomy) author statements anywhere here? <https://www.elsevier.com/authors/policies-and-guidelines/credit-author-statement>

3. Open data access as a public goods game

L134-135 - Unsure of why a sentence is contained with parentheses. Suggest: "Open data [7-11] is when research data are made publicly accessible at least at the time of first publication (ideally made accessible independently of publication, to avoid the file-drawer effect [12]).

L148-151 - Would be good to have some references to PGG for readers to investigate and refer to here and in the following paragraphs (e.g. from here up until L192).

L161-180 - I like this explanation of PGG - helps a lot.

4. Data contribution statements as a supplementary currency

L264-265 - And there is also variable interpretation of first authors versus last authors. And of course intermediary authors may also make major contributions to the collection, design, or analysis of data or the study, but are not proportionately recognised.

L269-270 – Suggest “What we suggest is that for job, promotion, and grant applications” to remove repeated ‘and’s.

Box 2 - I agree this would be a worthwhile addition to recognise contributions.

L293-295 – I’d point you to a recent preprint (I disclose by two researchers in my research group) that are proposing a new author-level metric that values impactful data and incentivises data sharing. <https://doi.org/10.1101/2020.10.20.344226>

L319-320 – Suggest “The result is a disseminated community opinion operating across these very many assessments” to remove splicing comma and lack of flow.

Review form: Reviewer 2

Recommendation

Major revision is needed (please make suggestions in comments)

Scientific importance: Is the manuscript an original and important contribution to its field?

Marginal

General interest: Is the paper of sufficient general interest?

Acceptable

Quality of the paper: Is the overall quality of the paper suitable?

Marginal

Is the length of the paper justified?

No

Should the paper be seen by a specialist statistical reviewer?

No

Do you have any concerns about statistical analyses in this paper? If so, please specify them explicitly in your report.

No

It is a condition of publication that authors make their supporting data, code and materials available - either as supplementary material or hosted in an external repository. Please rate, if applicable, the supporting data on the following criteria.

Is it accessible?

N/A

Is it clear?

N/A

Is it adequate?

N/A

Do you have any ethical concerns with this paper?

No

Comments to the Author

In this paper, the authors argue for a need to introduce a new currency for career progression based on data contributions, and suggest potential elements of a data contribution statement. While I find their idea interesting, and potentially relevant to the ongoing debate on how to encourage and officially recognize data sharing, I also have a few major comments about this work. I would generally suggest to shorten this MS into a comment/perspective. Further, this work is not particularly focused on the field of biology, so in my opinion, it could gain a better visibility in a more interdisciplinary journal.

While the authors try to develop the open data access as a public goods game, this model is not very well described, and no formal modelling has been done. This is an additional reason why I think this MS would work better as a comments - that would just describe the idea of how could data contribution be valued, and how could that improve data sharing (and hopefully data quality)

I also found that many of the statements within the MS are not backed up by any references (while I am aware that references exist). Further, the ongoing efforts (both top down and bottom up) to increase data sharing and the FAIRness of data have not been systematically (or fully) presented. I would also suggest some reference to the importance of FAIR (and not necessary open) data.

Finally, I am also unsure about how this data contribution statements would work. E.g. what with the data that have been collected within a large grant, funded by a PI, but collected and managed by Postdocs and PhDs? What about consciously collected long-term data where several generations of scientist participated in data collection and curation. Or what with the data collected by a person A who is no longer active, but hosted at an institution under a person B, who did not have anything with the data collection but is securing data access...

Decision letter (RSPB-2020-2830.R0)

08-Jan-2021

Dear Dr Westoby:

Your manuscript has now been peer reviewed and the reviews have been assessed by an Associate Editor. The reviewers' comments (not including confidential comments to the Editor) and the comments from the Associate Editor are included at the end of this email for your reference. As you will see, the reviewers and the Editors have raised some concerns with your manuscript and we would like to invite you to revise your manuscript to address them.

When submitting your revision please upload a file under "Response to Referees" - in the "File Upload" section. This should document, point by point, how you have responded to the

reviewers' and Editors' comments, and the adjustments you have made to the manuscript. We require a copy of the manuscript with revisions made since the previous version marked as 'tracked changes' to be included in the 'response to referees' document.

Research ethics:

Use of animals and field studies:

It is a condition of publication that you make available the data and research materials supporting the results in the article. Please see our Data Sharing Policies (<https://royalsociety.org/journals/authors/author-guidelines/#data>). Datasets should be deposited in an appropriate publicly available repository and details of the associated accession number, link or DOI to the datasets must be included in the Data Accessibility section of the article (<https://royalsociety.org/journals/ethics-policies/data-sharing-mining/>). Reference(s) to datasets should also be included in the reference list of the article with DOIs (where available).

Online supplementary material will also carry the title and description provided during submission, so please ensure these are accurate and informative. Note that the Royal Society will not edit or typeset supplementary material and it will be hosted as provided. Please ensure that

the supplementary material includes the paper details (authors, title, journal name, article DOI). Your article DOI will be 10.1098/rspb.[paper ID in form xxxx.xxxx e.g. 10.1098/rspb.2016.0049].

Please submit a copy of your revised paper within three weeks. If we do not hear from you within this time your manuscript will be rejected. If you are unable to meet this deadline please let us know as soon as possible, as we may be able to grant a short extension.

Best wishes,
Dr Maurine Neiman
mailto: proceedingsb@royalsociety.org

Associate Editor

Comments to Author:

I have now received the comments of two reviewers on your manuscript entitled “motivating data contributions via a distinct career currency”. Both reviewers thought the paper to be of broad interest, but they also had some major and minor suggestions for improvement. I hope you will find their comments useful; please consider them carefully when revising your paper. First, both reviewers thought that you do not sufficiently situate your ideas - please extend by giving more context and also provide more references (reviewer 1 made several good suggestions here). Second, both reviewers thought that the Public Good Games model is not described sufficiently enough, and reviewer 2 therefore even suggested to perhaps skip it and shorten the text to a short comment. However, reviewer 1 appreciated the descriptive model, and I do agree it is interesting. So please leave it in, but add a conceptual figure or diagram, as suggested by reviewer 1 (as I also agree that a formal model is not strictly necessary here). Third, reviewer 2 made an important final comment regarding the practicalities of how to assign data contributions to individual researchers; please address how you would solve this issue (or at least make the problem obvious and discuss it sufficiently). Finally, please address the minor quibbles (e.g. capitalization of titles etc).

Reviewer(s)' Comments to Author:

Referee: 1

Comments to the Author(s)

Review - motivating data contributions via a distinct career currency

General comments

I think this is a useful contribution to the debate about how to value researcher contributions and I like how the authors link the issue to Public Goods Games. I like the examples given to demonstrate the principles of PGGs and how they apply to issues surrounding open data, but maybe a diagram or conceptual figure could help to bring the article alive a little more? In places the grammar and sentence structure needs to be cleared up and I've made some comments below.

I'd also suggest some greater engagement with the literature on valuing researcher contributions - the article just generally feels a little light on references. I'd point the authors to a recent preprint (I disclose by two researchers in my research group) that are proposing a new author-level metric that values impactful data and incentivises data sharing: 'The data-index: an author-level metric that values impactful data and incentivises data sharing'

<https://doi.org/10.1101/2020.10.20.344226>. It would be interesting to discuss the strengths and limitations of such an approach here as the authors seem to err on the side of providing a qualitative assessment of data contributions in applications, rather than a quantitative metric to directly provide an alternative to the h-index and other authorship-related metrics.

Some more papers that could also be integrated are:

- ‘Transparency in authors’ contributions and responsibilities to promote integrity in scientific publication’ <https://doi.org/10.1073/pnas.1715374115>
- ‘Author contributions to ecological publications: What does it mean to be an author in modern ecological research?’ <https://journals.plos.org/plosone/article?id=10.1371/journal.pone.0179956>
- ‘Author sequence and credit for contributions in multiauthored publications’ <https://journals.plos.org/plosbiology/article?id=10.1371/journal.pbio.0050018>
- ‘Transparent Attribution of Contributions to Research: Aligning Guidelines to Real-Life Practices’ <https://doi.org/10.3390/publications7020024>
- ‘Attitudes to the rights and rewards for author contributions to repositories for teaching and learning’ <https://doi.org/10.1080/09687760600837066>
- ‘The need to quantify authors’ relative intellectual contributions in a multi-author paper’ <https://doi.org/10.1016/j.joi.2017.01.002>
- ‘Is the “academic conservation scientist” becoming an endangered species?’ <https://doi.org/10.1007/s13412-020-00633-6>.

Title

Title should be capitalised.

Abstract

The final sentence of the abstract (L21-24) has an erroneous or misplaced comma, it doesn’t flow.

Introduction

L36 – This sentence is missing a word. Should it be “researchers who have collected data make decisions ABOUT how much of it to make openly available”.

L38 – Same problem as for L36.

L35-43 – I wonder if there is any evidence or theories from psychology that can be used to strengthen this paragraph’s argument?

2. motivation via publications

Subheading should be capitalised.

L59 – Should be “Currencies can ALSO be thought of as motivations.”

L66 – ICMJE and ESA acronyms have not been defined yet (I know they are in Box 1 but that comes later). A reference to Box 1 should also be made so that people can understand what the protocols are if they don’t know them.

L66-67 – ‘Many data synthesis papers are being published with 10 or more authors’. Can you provide any evidence of this from the literature? Like you do on L138 for global data syntheses.

L70-72 – I agree! I myself have just published a paper where I included many co-authors purely for data contribution, and although I tried to get co-authors to contribute to the design and writing of the analysis, it’s inevitable that some contribute a great deal and some contribute extremely little.

L71 – Provide a link to the database?

I don't see any mention of the CRediT (Contributor Roles Taxonomy) author statements anywhere here? <https://www.elsevier.com/authors/policies-and-guidelines/credit-author-statement>

3. Open data access as a public goods game

L134-135 – Unsure of why a sentence is contained with parentheses. Suggest: “Open data [7–11] is when research data are made publicly accessible at least at the time of first publication (ideally made accessible independently of publication, to avoid the file-drawer effect [12]).

L148-151 – Would be good to have some references to PGG for readers to investigate and refer to here and in the following paragraphs (e.g. from here up until L192).

L161-180 – I like this explanation of PGG – helps a lot.

4. Data contribution statements as a supplementary currency

L264-265 – And there is also variable interpretation of first authors versus last authors. And of course intermediary authors may also make major contributions to the collection, design, or analysis of data or the study, but are not proportionately recognised.

L269-270 – Suggest “What we suggest is that for job, promotion, and grant applications” to remove repeated ‘and’s.

Box 2 - I agree this would be a worthwhile addition to recognise contributions.

L293-295 – I'd point you to a recent preprint (I disclose by two researchers in my research group) that are proposing a new author-level metric that values impactful data and incentivises data sharing. <https://doi.org/10.1101/2020.10.20.344226>

L319-320 – Suggest “The result is a disseminated community opinion operating across these very many assessments” to remove splicing comma and lack of flow.

Referee: 2

Comments to the Author(s)

In this paper, the authors argue for a need to introduce a new currency for career progression based on data contributions, and suggest potential elements of a data contribution statement. While I find their idea interesting, and potentially relevant to the ongoing debate on how to encourage and officially recognize data sharing, I also have a few major comments about this work. I would generally suggest to shorten this MS into a comment/perspective. Further, this work is not particularly focused on the field of biology, so in my opinion, it could gain a better visibility in a more interdisciplinary journal.

While the authors try to develop the open data access as a public goods game, this model is not very well described, and no formal modelling has been done. This is an additional reason why I think this MS would work better as a comments – that would just describe the idea of how could data contribution be valued, and how could that improve data sharing (and hopefully data quality)

I also found that many of the statements within the MS are not backed up by any references (while I am aware that references exist). Further, the ongoing efforts (both top down and bottom up) to increase data sharing and the FAIRness of data have not been systematically (or fully) presented. I would also suggest some reference to the importance of FAIR (and not necessary open) data.

Finally, I am also unsure about how this data contribution statements would work. E.g. what with the data that have been collected within a large grant, funded by a PI, but collected and managed by Postdocs and PhDs? What about consciously collected long-term data where several generations of scientist participated in data collection and curation. Or what with the data collected by a person A who is no longer active, but hosted at an institution under a person B, who did not have anything with the data collection but is securing data access...

Author's Response to Decision Letter for (RSPB-2020-2830.R0)

See Appendix A.

Decision letter (RSPB-2020-2830.R1)

01-Feb-2021

Dear Dr Westoby

I am pleased to inform you that your Review manuscript RSPB-2020-2830.R1 entitled "motivating data contributions via a distinct career currency" has been accepted for publication in Proceedings B.

The referee(s) do not recommend any further changes. Therefore, please proof-read your manuscript carefully and upload your final files for publication. Because the schedule for publication is very tight, it is a condition of publication that you submit the revised version of your manuscript within 7 days. If you do not think you will be able to meet this date please let me know immediately.

To upload your manuscript, log into <http://mc.manuscriptcentral.com/prsb> and enter your Author Centre, where you will find your manuscript title listed under "Manuscripts with Decisions." Under "Actions," click on "Create a Revision." Your manuscript number has been appended to denote a revision.

You will be unable to make your revisions on the originally submitted version of the manuscript. Instead, upload a new version through your Author Centre.

- 1) A text file of the manuscript (doc, txt, rtf or tex), including the references, tables (including captions) and figure captions. Please remove any tracked changes from the text before submission. PDF files are not an accepted format for the "Main Document".
- 2) A separate electronic file of each figure (tiff, EPS or print-quality PDF preferred). The format should be produced directly from original creation package, or original software format. Please note that PowerPoint files are not accepted.
- 3) Electronic supplementary material: this should be contained in a separate file from the main text and the file name should contain the author's name and journal name, e.g. `authorname_procb_ESM_figures.pdf`

All supplementary materials accompanying an accepted article will be treated as in their final form. They will be published alongside the paper on the journal website and posted on the online figshare repository. Files on figshare will be made available approximately one week before the accompanying article so that the supplementary material can be attributed a unique DOI. Please see: <https://royalsociety.org/journals/authors/author-guidelines/>

4) Data-Sharing and data citation

It is a condition of publication that data supporting your paper are made available. Data should be made available either in the electronic supplementary material or through an appropriate repository. Details of how to access data should be included in your paper. Please see <https://royalsociety.org/journals/ethics-policies/data-sharing-mining/> for more details.

If you wish to submit your data to Dryad (<http://datadryad.org/>) and have not already done so you can submit your data via this link <http://datadryad.org/submit?journalID=RSPB&manu=RSPB-2020-2830.R1> which will take you to your unique entry in the Dryad repository.

Once again, thank you for submitting your manuscript to Proceedings B and I look forward to receiving your final version. If you have any questions at all, please do not hesitate to get in touch.

Sincerely,
Dr Maurine Neiman
Editor, Proceedings B
<mailto:proceedingsb@royalsociety.org>

Associate Editor Board Member
Comments to Author:

I think you have done a very good job with this revision, and the new figure provides a nice schematic overview now. But could you please re-check the figure as it seems to contain several minor mistakes:

- You should replace "freeloaders" with "free-riders"
- The statement in the green box seems odd; I would suggest a change to "Increased motivation to collect and contribute data driven by peer valuation"
- Add capitalisation where needed (in at least two places)

Decision letter (RSPB-2020-2830.R2)

02-Feb-2021

Dear Dr Westoby

I am pleased to inform you that your manuscript entitled "motivating data contributions via a distinct career currency" has been accepted for publication in Proceedings B.

Your article has been estimated as being 6 pages long. Our Production Office will be able to confirm the exact length at proof stage.

Open Access

Paper charges

Sincerely,

Appendix A

response to refs 2021-1-27

Thank you for these constructive reviews. Particular thanks also to the Associate Editor for giving such clear recommendations, where there is a choice to be made. We believe the revision is substantially strengthened as a result.

A complete version of the revised ms with track changes is attached at the end of this response to referee comments

Associate Editor

Comments to Author:

I have now received the comments of two reviewers on your manuscript entitled “motivating data contributions via a distinct career currency”. Both reviewers thought the paper to be of broad interest, but they also had some major and minor suggestions for improvement. I hope you will find their comments useful; please consider them carefully when revising your paper. First, both reviewers thought that you do not sufficiently situate your ideas - please extend by giving more context and also provide more references (reviewer 1 made several good suggestions here). Second, both reviewers thought that the Public Good Games model is not described sufficiently enough, and reviewer 2 therefore even suggested to perhaps skip it and shorten the text to a short comment. However, reviewer 1 appreciated the descriptive model, and I do agree it is interesting. So please leave it in, but add a conceptual figure or diagram, as suggested by reviewer 1 (as I also agree that a formal model is not strictly necessary here). Third, reviewer 2 made an important final comment regarding the practicalities of how to assign data contributions to individual researchers; please address how you would solve this issue (or at least make the problem obvious and discuss it sufficiently). Finally, please address the minor quibbles (e.g. capitalization of titles etc).

Mainly the revision makes changes (and we hope strengthens) in two areas, and we would like to comment on them here.

One is the question how our proposal for a free-form data contributions section should be compared to indices that quantify data contributions. In fact, we think quantitative indices would often be included within a free-form statement. Yet at the same time indices can often be fairly misleading if taken without explanation. They work better if complemented with a qualitative account about types of contributions made and their direction. The revision addresses this topic directly and provides expanded explanation in association with Box 2.

Second is the matter of the CRediT taxonomy of author contributions together with the FAIR principles for data access. We agree these should both be mentioned. At the same time, neither of them address directly the question we are concerned with, which is motivation for data contributions. And we feel it would be unhelpful for this essay inflate to become a general discourse about authorship, data access principles and so forth. Many others have written on those topics. The distinctive features of our ms are the PGG interpretation of motivations, and the concrete and workable proposal for data contributions statements. So CRediT and FAIR are mentioned in this revision, but they are not discussed at length.

Reviewer(s)' Comments to Author:

Referee: 1

Comments to the Author(s)

Review - motivating data contributions via a distinct career currency

General comments

I think this is a useful contribution to the debate about how to value researcher contributions and I like how the authors link the issue to Public Goods Games. I like the examples given to demonstrate the principles of PGGs and how they apply to issues surrounding open data, but maybe a diagram or conceptual figure could help to bring the article alive a little more? In places the grammar and sentence structure needs to be cleared up and I've made some comments below.

Thanks for the encouragement!

We have provided a schematic as Fig 1. Its three sectors correspond to how the text proceeds through the PGG framing – left hand side is the basic PGG, across the top are processes that mitigate the expected free-loading, right hand side is proposed solutions via data contributions statements and indices.

I'd also suggest some greater engagement with the literature on valuing researcher contributions – the article just generally feels a little light on references. I'd point the authors to a recent preprint (I disclose by two researchers in my research group) that are proposing a new author-level metric that values impactful data and incentivises data sharing: 'The data-index: an author-level metric that values impactful data and incentivises data sharing' <https://doi.org/10.1101/2020.10.20.344226>. It would be interesting to discuss the strengths and limitations of such an approach here as the authors seem to err on the side of providing a qualitative assessment of data contributions in applications, rather than a quantitative metric to directly provide an alternative to the h-index and other authorship-related metrics.

Many thanks for pointing us to more literature, and especially to this very interesting Hood and Sutherland preprint in BioRxiv. It addresses (we think) exactly the same problem as we are concerned with, that the pubs-and-cites credit system cannot satisfactorily motivate data contributions. Their proposed index is a different solution from what we propose, but we see the two solutions as complementary rather than competing.

It is true that we advocate a free-form presentation of data contributions. But we do not mean to reject indices such as the data-index that Hood and Sutherland propose. We fully expect that this or other indices would come to be used within the data contributions statements that we propose. But at the same time, we believe that any one index necessarily has limitations. For example, there are the complications adduced by ref 2 about who should actually get the credit for a given body of data. And it's well known that once an index becomes widely adopted, it also becomes gamed. Overall, experience with publication-credit suggests that providing for multiple indices plus for qualitative explanation makes possible the fairest overall assessment.

Some more papers that could also be integrated are:

- 'Transparency in authors' contributions and responsibilities to promote integrity in scientific publication' <https://doi.org/10.1073/pnas.1715374115>

- ‘Author contributions to ecological publications: What does it mean to be an author in modern ecological research?’

<https://journals.plos.org/plosone/article?id=10.1371/journal.pone.0179956>

- ‘Author sequence and credit for contributions in multiauthored publications’

<https://journals.plos.org/plosbiology/article?id=10.1371/journal.pbio.0050018>

- ‘Transparent Attribution of Contributions to Research: Aligning Guidelines to Real-Life Practices’ <https://doi.org/10.3390/publications7020024>

- ‘Attitudes to the rights and rewards for author contributions to repositories for teaching and learning’ <https://doi.org/10.1080/09687760600837066>

- ‘The need to quantify authors’ relative intellectual contributions in a multi-author paper’ <https://doi.org/10.1016/j.joi.2017.01.002>

- ‘Is the “academic conservation scientist” becoming an endangered species?’ <https://doi.org/10.1007/s13412-020-00633-6>.

Many thanks for these pointers. They are interesting and several are now cited.

Title

Title should be capitalised.

done

Abstract

The final sentence of the abstract (L21-24) has an erroneous or misplaced comma, it doesn’t flow.

The passage has been re-cast

Introduction

L36 – This sentence is missing a word. Should it be “researchers who have collected data make decisions ABOUT how much of it to make openly available”.

done

L38 – Same problem as for L36.

done

L35-43 – I wonder if there is any evidence or theories from psychology that can be used to strengthen this paragraph’s argument?

This para is just saying that researchers make decisions how to allocate their time, and that they are motivated by advancing their career (among other motivations, of course). This much is surely common understanding in the research world?

We are not persuaded this argument would be strengthened by linking it to a theory in psychology. For example, Maslow's hierarchy of needs is probably the most widely-known theory of motivation. Starting from the bottom of the hierarchy, success in a research career satisfies first physiology and safety (via getting paid), then social needs, then esteem, then self-actualisation. But in all honesty, we don't think that couching the point in that way is going to make it more persuasive to readers.

2. motivation via publications

Subheading should be capitalised.

done

L59 – Should be “Currencies can ALSO be thought of as motivations.”

done

L66 – ICMJE and ESA acronyms have not been defined yet (I know they are in Box 1 but that comes later). A reference to Box 1 should also be made so that people can understand what the protocols are if they don't know them.

There is reference to Box 1 at the beginning of the para. But in addition, ICMJE and ESA have now been spelled out in main text.

L66-67 – ‘Many data synthesis papers are being published with 10 or more authors’. Can you provide any evidence of this from the literature? Like you do on L138 for global data syntheses.

Some examples have been provided.

L70-72 – I agree! I myself have just published a paper where I included many co-authors purely for data contribution, and although I tried to get co-authors to contribute to the design and writing of the analysis, it's inevitable that some contribute a great deal and some contribute extremely little.

L71 – Provide a link to the database?

done

I don't see any mention of the CRediT (Contributor Roles Taxonomy) author statements anywhere here? <https://www.elsevier.com/authors/policies-and-guidelines/credit-author-statement>

CRediT is now described, mainly in an introductory paragraph within Box 1.

3. Open data access as a public goods game

L134-135 – Unsure of why a sentence is contained with parentheses. Suggest: “Open data [7–11] is when research data are made publicly accessible at least at the time of first publication (ideally made accessible independently of publication, to avoid the file-drawer effect [12]).

We have adopted the style preferred by ref 1.

L148-151 – Would be good to have some references to PGG for readers to investigate and refer to here and in the following paragraphs (e.g. from here up until L192).

In our judgment the clearest short overview of public goods games is in Wikipedia, so that is what we have cited. Zotero has elected to cite this as an encyclopaedia entry. We could force it to be cited as a website if that was preferred.

L161-180 – I like this explanation of PGG – helps a lot.

Thanks!

4. Data contribution statements as a supplementary currency

L264-265 – And there is also variable interpretation of first authors versus last authors. And of course intermediary authors may also make major contributions to the collection, design, or analysis of data or the study, but are not proportionately recognised.

Indeed. And there has been much discussion of how practices vary between disciplines and countries, and a wide range of proposals to standardize author participation or author order or to construct indices that somehow capture different contributions. All this has brought us to the view put forward in this ms, that authorship is never going to capture data contributions in a satisfactory way. Partly because of all these complications, and partly because of the conflicting aim that authorship should reflect taking responsibility for analysis and conclusions.

L269-270 – Suggest “What we suggest is that for job, promotion, and grant applications” to remove repeated ‘and’s.

The style preferred by ref 1 has been adopted.

Box 2 - I agree this would be a worthwhile addition to recognise contributions.

Pleased to hear it – it is the central practical recommendation.

L293-295 – I’d point you to a recent preprint (I disclose by two researchers in my research group) that are proposing a new author-level metric that values impactful data and incentivises data sharing. <https://doi.org/10.1101/2020.10.20.344226>

Many thanks for this interesting pointer. As well as including reference to it within Box 2, we have added text in the discussion of box 2 to make clear that the data contributions statements we recommend will very commonly include quantitative indices such as the one proposed in this preprint. We see these as complementary rather than conflicting ideas.

L319-320 – Suggest “The result is a disseminated community opinion operating across these very many assessments” to remove splicing comma and lack of flow.

The sentence has been rewritten. We would like to keep the emphasis on spread – on the idea that in aggregate this will be a fair expression of the judgment of the community of research, though of course individual decisions can be more or less misguided.

Referee: 2

Comments to the Author(s)

In this paper, the authors argue for a need to introduce a new currency for career progression based on data contributions, and suggest potential elements of a data contribution statement. While I find their idea interesting, and potentially relevant to the ongoing debate on how to encourage and officially recognize data sharing, I also have a few major comments about this work. I would generally suggest to shorten this MS into a comment/perspective. Further, this work is not particularly focused on the field of biology, so in my opinion, it could gain a better visibility in a more interdisciplinary journal.

We appreciate ref 2's point of view. However, on balance we have gone with the Assoc Editor's recommendation to flesh out the paper more so than to prune it.

While the authors try to develop the open data access as a public goods game, this model is not very well described, and no formal modelling has been done. This is an additional reason why I think this MS would work better as a comments – that would just describe the idea of how could data contribution be valued, and how could that improve data sharing (and hopefully data quality)

We are sorry ref 2 did not find the application of PGG helpful. We think it sheds light on conflicting motivations from different participants. Much of the campaign for open data seems to have taken it for granted that open data serves the general good – that we are only held back from achieving fully open data because people are a little lazy or a little out of date.

Ref 2's point about data quality is important, we think. It is one of the reasons for advocating free-form data contributions statements, not relying only on quantitative indices.

I also found that many of the statements within the MS are not backed up by any references (while I am aware that references exist). Further, the ongoing efforts (both top down and bottom up) to increase data sharing and the FAIRness of data have not been systematically (or fully) presented. I would also suggest some reference to the importance of FAIR (and not necessary open) data.

Reference to FAIR has been added (first para of section 3). Further citations have been added to the text more broadly. At the same time, this paper is not aiming to be a general review of issues in data sharing.

Finally, I am also unsure about how this data contribution statements would work. E.g. what with the data that have been collected within a large grant, funded by a PI, but collected and managed by Postdocs and PhDs? What about consciously collected long-term data where several generations of scientist participated in data collection and curation. Or what with the data collected by a person A who is no longer active, but hosted at an institution under a person B, who did not have anything with the data collection but is securing data access...

Because of complications of this kind, we believe that even after consistent quantitative indices for data contributions come to be widespread, they would never be satisfactory by themselves. This is the reason for inviting a free-form statement of data contributions. Of course, people writing such statements can be expected to put a positive spin on their personal role. But they would be aware that referees or selection panel members might quite likely include people who had knowledge of their own about the datasets or team projects. We have written an expanded section in association with Box 2 that enlarges on these matters.

Finally (and not in response to reviewer comments) we should note a small formatting matter we have not been able to solve. Two citations to websites in Box 1 are described in Lit Cit by our reference manager Zotero as “in press”, and the citations do not give the url or the date accessed, even though this information is available in the primary Zotero record. We do not know why Zotero is unsatisfactory in this way. We have given the urls in the main text. We trust this problem can be worked out somehow, in the event the manuscript is acceptable to RSPB.